

# Mapping the hidden diversity of the *Geophagus sensu stricto* species group (Cichlidae: Geophagini) from the Amazon basin

Aline Mourão Ximenes[1,2], Pedro Senna Bittencourt[1], Valéria Nogueira Machado[1], Tomas Hrbek[1,3] and Izeni Pires Farias[1]

[1] Departamento de Genética, Instituto de Ciências Biológicas, Universidade Federal do Amazonas, Manaus, Amazonas, Brazil
[2] Programa de Pós-Graduação em Genética, Conservação e Biologia Evolutiva, Instituto Nacional de Pesquisas da Amazônia, Manaus, Amazonas, Brazil
[3] Biology Department, Trinity University, San Antonio, Texas, USA

## ABSTRACT

South American freshwater ichthyofauna is taxonomically the most diverse on the planet, yet its diversity is still vastly underestimated. The Amazon basin alone holds more than half of this diversity. The evidence of this underestimation comes from the backlog of morphologically distinct, yet undescribed forms deposited in museum collections, and from DNA-based inventories which consistently identify large numbers of divergent lineages within even well-studied species groups. In the present study, we investigated lineage diversity within the *Geophagus sensu stricto* species group. To achieve these objectives, we analyzed 337 individuals sampled from 77 locations within and outside the Amazon basin representing 10 nominal and six morphologically distinct but undescribed species. We sequenced the mitochondrial cytochrome c oxidase subunit I (COI) and delimited lineages using four different single-locus species discovery methods (mPTP-15 lineages; LocMin-14 lineages; bGMYC-18 lineages; and GMYC-30 lineages). The six morphologically distinct but undescribed species were also delimited by the majority of the species discovery methods. Five of these lineages are restricted to a single collection site or a watershed and their habitats are threatened by human activities such as deforestation, agricultural activities and construction of hydroelectric plants. Our results also highlight the importance of combining DNA and morphological data in biodiversity assessment studies especially in taxonomically diverse tropical biotas.

# INTRODUCTION

South American freshwater ichthyofauna is the most diverse on the planet with more than 5,150 valid described species (*Reis et al., 2016*). The Amazon River basin alone holds more than half of this diversity (52%-2,716 valid species) (*Dagosta & De Pinna, 2019*), even

Corresponding authors
Aline Mourão Ximenes,
alineximenesbio@gmail.com
Izeni Pires Farias,
izeni@evoamazon.net

though extensive areas to the north and south of the main river channel still remain poorly sampled (*Reis et al., 2016*). The evolution of this astonishing diversity resulted from continent-wide geomorphological processes forming and reshaping the Amazon basin hydrological network beginning in the Miocene, and climatic oscillations beginning in the Pliocene (*Montoya-Burgos, 2003*; *Hubert & Renno, 2006*; *Lovejoy, Albert & Crampton, 2006*; *Reis et al., 2016*; *Bloom & Lovejoy, 2017*). These processes have resulted in not only vicariance and geodispersal of entire fish communities (*Dagosta & De Pinna, 2018*), but also have generated an environmentally, physico-chemically and structurally heterogeneous landscape (*Rodríquez et al., 2007*; *Gregory-Bogotá et al., 2020*). Therefore, environmental heterogeneity, climate, ecological and historical factors have an important role in explaining the current diversity of Amazonian fishes (*Oberdorff et al., 2019*).

Sequences of the mitochondrial DNA gene cytochrome c oxidase (COI) are often used to assist taxonomy and species identification following the DNA barcoding principles (*Hebert, Ratnasingham & DeWaard, 2003*; *Hajibabaei et al., 2007*). They can also be used for biodiversity inventories (*Monaghan et al., 2009*; *Carvalho et al., 2011*; *Pereira et al., 2013*; *Machado et al., 2018*), allowing rapid characterization of not just a given sample but of entire communities (*Carvalho et al., 2018*; *Machado et al., 2018*; *Souza et al., 2018*; *Arruda et al., 2019*; *Santos et al., 2019*). The simplest approach to delimit species from DNA sequence data is to use the percent cut-off rule (*Fujisawa & Barraclough, 2013*), such as the 2% intra *vs.* interspecific divergence cut-off for freshwater fishes (*Pereira et al., 2011*, *2013*). There are also methods that automatically optimize the cut-off percentage for a given sample (*e.g.*, ABGD, Locmin) (*Brown et al., 2012*; *Puillandre et al., 2012*). However, distance-based methods are weakly connected to evolutionary theory (*Fujisawa & Barraclough, 2013*), since they ignore evolutionary relationships of the taxa involved and rely on sequence similarity thresholds that are not necessarily biologically relevant (*Kapli et al., 2017*).

On the other hand, the pattern of evolutionary relationships of taxa accurately reflects the processes that resulted in the gene trees, consequently, permitting differentiation between intra and interspecific patterns of evolutionary relationship (*Fujisawa & Barraclough, 2013*). Several coalescent and movement-based algorithms capable of accurately differentiating between intra and inter specific patterns of phylogenetic relationships have been proposed (*e.g.*, *Reid & Carstens, 2012*; *Fujisawa & Barraclough, 2013*; *Zhang et al., 2013*; *Kapli et al., 2017*). Despite the limitations of inference from single locus data (*Fujisawa & Barraclough, 2013*; *Dellicour & Flot, 2018*), single locus species delimitation methods (SLSD) provide a robust framework for species discovery. They identify unique evolutionary histories even if conflicting with genomic data, and they often provide the initial hypothesis for the existence of new species within a given dataset stimulating taxonomic studies (*Ota et al., 2020*).

Recent DNA-barcoding inventories and the usage of SLSD methods indicate that for some fish families widely distributed across the Amazon basin (*e.g.*, Cichlidae and Serrasalmidae), the underestimation of diversity appears to be broadly concordant with *Reis et al. (2016)*, who estimated that 34–42% of Neotropical freshwater fishes remain undescribed. This is also consistent with the study of *Melo et al. (2021)*, who studied

patterns of diversification of Characoid fishes, and identified a burst of diversification in Anostomidae, Serrasalmidae and Characidae families. DNA-based species discovery analyses focusing on lower taxonomic units such as genera of cichlids, also discovered multiple divergent evolutionary lineages (*e.g., Astronotus* Swainson, 1839 (*Colatreli et al., 2012*); *Apistogramma* Regan, 1913 (*Tougard et al., 2017*); *Gymnogeophagus* Miranda-Ribeiro, 1918 (*Říčan et al., 2018*); *Australoheros* Říčan & Kullander, 2006 (*Ottoni et al., 2019*); and *Geophagus* Heckel, 1840 (*Alves-Silva & Dergam, 2014*; *Carvalho et al., 2018*; *Argolo et al., 2020*)) suggesting cryptic diversity and, possibly, undescribed species.

The cichlid genus *Geophagus* comprises 31 species of eartheaters (*Fricke, Eschmeyer & Van der Lann, 2020*) grouped into three species groups (*Kullander, 1998*; *López-Fernández & Taphorn, 2004*). These species groups came into use after *Kullander's (1986)* revision of the genus, which restricted *Geophagus* to include only species with paired caudal extensions of the swim bladder supported by epihemal ribs, and greater number of caudal than abdominal vertebrae–morphological features absent in the species of the 'G.' steindachneri and 'G.' brasiliensis species groups. The three groups also have allopatric distribution in the Neotropics: The *Geophagus sensu stricto* species group, with 20 species is distributed within the Amazon, Orinoco, Parnaiba, and northern Atlantic coast river basins; the 'Geophagus' *steindachneri* species group, with three trans-Andean species is distributed in Panamá, Colombia and Venezuela; and the 'Geophagus' *brasiliensis* species group, with eight species is distributed in eastern South American river basins. While the 'Geophagus' *steindachneri* and 'Geophagus' *brasiliensis* species groups have been left without a formal generic assignment (*Argolo et al., 2020*) and their phylogenetic relationships remains unclear (*López-Fernández, Winemiller & Honeycutt, 2010*; *Ilves, Torti & López-Fernández, 2018*), this has not impeded active taxonomic interest in these species groups. Recently an integrative taxonomic analysis of the 'G.' brasiliensis species group using SLSD methods, multilocus RADseq data, and geometric morphometrics (*Argolo et al., 2020*) provided support for the eight nominal species of this group and suggested the recognition of an additional two species, increasing the taxonomic diversity of this group by 20%.

The *Geophagus sensu stricto* species group has received relatively little recent taxonomic attention. *Geophagus sensu stricto* includes both broadly distributed species, such as *G. altifrons* Heckel, 1840 and *G. proximus* (Castelnau, 1855) and species with restricted distributions such as *G. mirabilis* Deprá et al., 2014 endemic to the upper Aripuanã River (upstream of the Dardanelos/Andorinhas rapids) and *G. argyrostictus* Kullander, 1991 occuring in the Belo Monte rapids region of the Xingu River. Many of these species were previously referred to as *G. surinamensis* Bloch, 1791 (*Regan, 1906*), but currently these species are part of the *G. surinamensis* complex (except *G. argyrostictus*) which contains an undetermined number of undescribed species distributed in the Orinoco and Amazon basins (*López-Fernández & Taphorn, 2004*). Several putative species of *Geophagus* have been proposed for the Amazon basin as well (*López-Fernández & Taphorn, 2004*; *Ohara et al., 2017*; *Oliveira et al., 2020*).

Due to known taxonomic uncertainties in the genus *Geophagus* (*López-Fernández & Taphorn, 2004*; *Carvalho et al., 2018*; *Argolo et al., 2020*), we produced a COI sequence
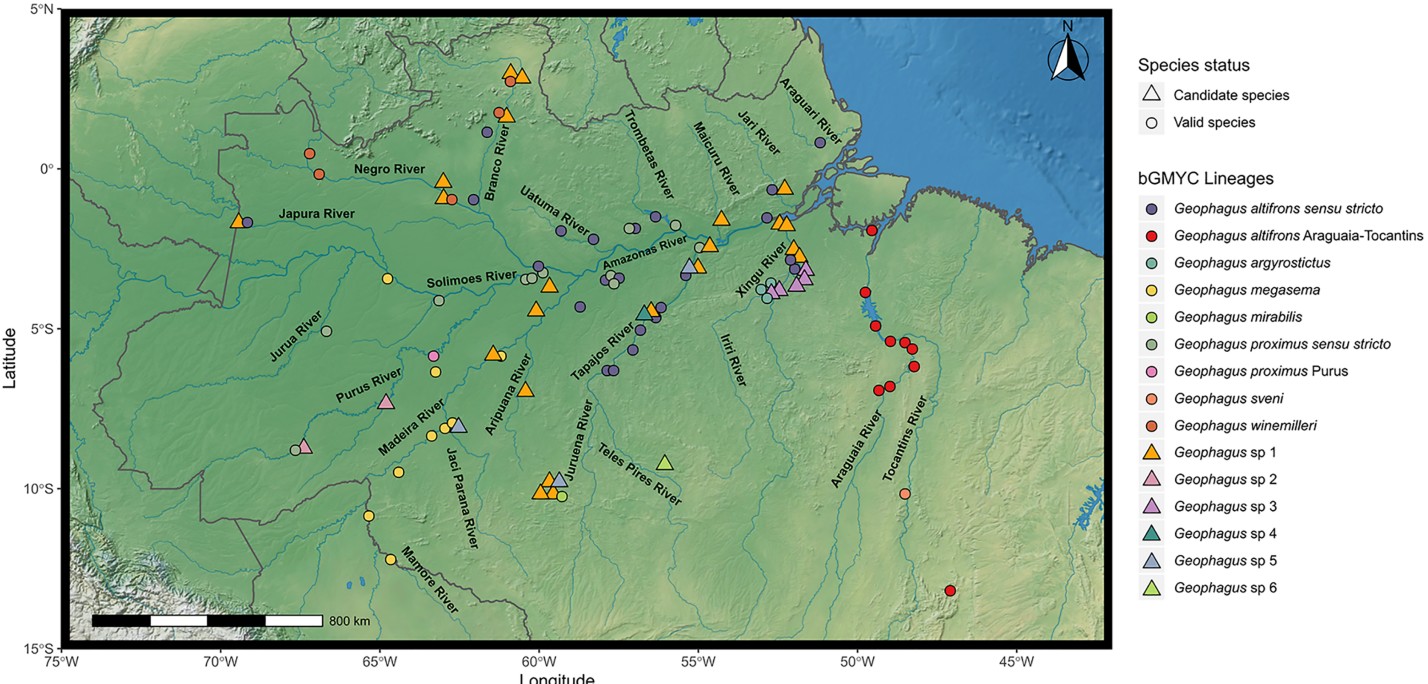

**Figure 1** Map showing all the sites sampled in this study, including data accessed from GenBank and BOLD databases when available. A total of 77 localities were sampled. A small amount of random variation to the location of each point was added to prevent overplotting. The symbols refer to the morpho-species identified *a priori* in this study, and the colors refer to the lineages found in bGMYC analysis. The map was constructed in R 4.0.0 using packages 'ggspatial', 'raster', 'rgdal', 'rnaturalearth', and 'tidyverse'. The final image was edited in Inkscape.

dataset for the species of the *Geophagus sensu stricto* species group from the Brazilian Amazon, with the aim (1) to investigate lineage diversity, and (2) to identify species complexes and their distribution patterns. We use the genealogical phylogenetic species concept (*Baum & Shaw, 1995*) to identify lineages (*De Queiroz, 2007*), and posteriorly we test their morphological distinctness.

## MATERIALS & METHODS

### Sampling

A total of 315 individuals of *Geophagus* were sampled from 72 localities within the Amazon River basin (Fig. 1; Table S1). For each specimen, we collected the right pectoral fin or muscle tissue and preserved it in 96% ethanol for further laboratory analyses. Vouchers were preserved in 10% formaldehyde, and subsequently transferred to 70% ethanol for long-term storage. We obtained samples during field expeditions, from fishermen, local markets, or through tissue collections of the Universidade Federal de Rondônia (UNIR); the Ichthyological Collection of Instituto Nacional de Pesquisas da Amazônia (INPA); and from Animal Genetics Tissue Collection of the Laboratório de Evolução e Genética Animal of the Universidade Federal do Amazonas (CTGA-UFAM). All individuals were captured and sampled under license granted by the Instituto Brasileiro do Meio Ambiente e dos Recursos Naturais Renováveis (IBAMA/SISBIO permit #62216-1). Collection of organisms was undertaken in accordance with the ethical

recommendations of the Conselho Federal de Biologia (CFBio; Federal Council of Biologists), Resolution 301 (December 8, 2012).

## Morphological identification

For the morphological identification of *Geophagus* individuals, we used original descriptions and identification keys (*Kullander, 1991*; *López-Fernández & Taphorn, 2004*; *Lucinda, Lucena & Assis, 2010*; *Deprá et al., 2014*). Thus, identified individuals are hereinafter referred to as morpho-species. For those individuals that could not be identified to the species level using the original descriptions and taxonomic keys, the following nomenclature was used: *Geophagus* sp. (possible new/unidentified species) (*Bengtson, 1988*; *Sigovini, Keppel & Tagliapietra, 2016*).

## COI sequence data generation

Genomic DNA was isolated using the phenol-chloroform method (*Sambrook, Fritsch & Maniatis, 1989*). DNA integrity was visualized on a 0.8% agarose gel stained with GelRed (Biotium). Quantification and quality of DNA were checked spectrophotometrically in Nanodrop 2000 (Thermo-Scientific) and diluted to a final concentration of 50 ng/μl.

The partial fragment of mitochondrial cytochrome c oxidase subunit I (COI) was amplified in a 15 μl PCR mix containing: 7.6 μl of ddH$_2$O, 1.2 μl dNTP (10 mM), 1.5 μl buffer 10X (100 mM Tris-HCl, 500 mM KCl), 1.2 μl MgCl$_2$ (25 mM), 1.2 μl of primers COI-Fish-f.2 and COI-Fish-r.1 (2 pM each) (*Ivanova et al., 2007*), 0.5μl of Taq DNA polymerase (1 U/μl) and 1μL of template DNA (final concentration of 50 ng/μl). PCR cycling conditions were as follows: denaturation at 93 °C for 1 min, 35 cycles of denaturation at 93 °C for 10 s, annealing at 50 °C for 45 s, and extension at 72 °C for 1 min, followed by a final extension cycle of 72 °C for 7 min. PCR products were purified using ExoSap and subjected to fluorescent dye-terminator (ddNTP) sequencing following the manufacturer's recommended protocol for BigDye sequencing chemistry (Applied Biosystems). Purified amplicons were sequenced on an automatic ABI 3500 sequencer (Applied Biosystems).

The organization, verification, and edition of the sequences were carried out in Geneious software v 7.0.6 (*Kearse et al., 2012*). The chromatogram reads for each sample sequenced were assembled into contigs and verified visually. We also translated the contigs into putative amino acids to check for the presence of stop codons; no internal stop codons were found. The alignment tool MAFFT v7.307 (*Katoh & Standley, 2013*) was used to perform the alignment which was later edited manually. Twenty two GenBank and BOLD sequences (accession number: *Geophagus proximus*-FUPR931-09, FUPR932-09, FUPR933-09, FUPR934-09, FUPR935-09, GU701784; *G. sveni*-MH780911, MK012088; *G. harreri*-DSFRE369-08; *G. argyrostictus*-PARO177-08, PARO178-08, PARO179-08, PARO180-08; *G. surinamensis*-KU568829, KU568830, DSFRE196-08, BNAF153-09, BNAF152-09; *G. dicrozoster*-DSFRE170-08, DSFRE171-08, DSFRE138-08; '*Geophagus*' *steindachneri*-KR150866) were added to the alignment, which added five sampling sites outside the Amazon basin, representing four localities in the Paraná River basin and one in the upper Tocantins River. '*Geophagus*' *steindachneri* (KR150866 and CTGA 145) was

used as an outgroup based in the phylogenetic relationship of Geophagines (*López-Fernández, Honeycutt & Winemiller, 2005*). Thus, the final dataset comprised 337 individuals sampled from 77 localities. A Neighbor-Joining tree containing all sequences is provided as Supplemental material (Fig. S1). All new sequences generated in this study are available in GenBank under accessions MZ504295–MZ504609. Metadata for all sequences used in this study are presented in Table S1 as a flat file following the standard Darwin core format (http://rs.tdwg.org/dwc/terms/index.htm).

## Species discovery analyses

For single-locus species discovery (SLSD) analyses, the total dataset was reduced to a new dataset containing unique haplotypes using the hapCollapse function (available at http://github.com/legalLab/protocols-scripts) in the statistical software R (*R Development Core Team, 2011*). We then generated a Bayesian Inference phylogeny using the software BEAST 2.6.2 (*Bouckaert et al., 2019*) using the following settings: nucleotide substitution model (TrN + I + G) estimated using the BEAST2 package bModelTest 1.2.1 (*Bouckaert & Drummond, 2017*); single site model partition; strict molecular clock; Yule model tree prior. We ran three independent runs with 20 million Markov Chain Monte Carlo (MCMC) generations each, sampling tree topologies and parameters every 2,000 generations. The convergence of parameters of each run was observed by checking the values of effective sample size (ESS > 200) and stationarity of the chain using the software TRACER 1.7.1 (*Rambaut et al., 2018*). We combined the runs, subsampled at a frequency of 6,000 generations, and burned-in the first 10% generations of each run using LogCombiner (*Drummond et al., 2012*) to produce a final dataset with 9,000 topologies which were used to produce a maximum credibility tree in TREEANNOTATOR (*Bouckaert et al., 2019*).

We used the maximum credibility tree as input for four single-locus species discovery analyses: GMYC, the Generalized Mixed Yule Coalescent method (*Fujisawa & Barraclough, 2013*); bGMYC, a Bayesian implementation of GMYC (*Reid & Carstens, 2012*); mPTP, the multi-rate Poisson tree process method (*Kapli et al., 2017*); and LocMin, a threshold distance based method (*Brown et al., 2012*). For GMYC, we used the package splits_1.0–19 (*Fujisawa & Barraclough, 2013*); for bGMYC, we used the package bGMYC 1.0.2 (*Reid & Carstens, 2012*). For mPTP, we transformed the BEAST tree into a rooted phylogram using the 'optim.pml' function of phangorn_2.3.1 (*Schliep, 2011*), optimizing the topology, branch lengths, and gamma rate parameters. The phylogram was used as input into the stand-alone software mptp 0.2.3 (*Kapli et al., 2017*). We also used a p-distance based method using the 'locMin' and 'tclust' functions, a distance threshold optimization and a clustering approach implemented in SPIDER (*Brown et al., 2012*). All analyses were carried out in the R statistical software v. 3.6.2 (*R Development Core Team, 2011*) and visualized using the package ggtree (*Yu et al., 2017*).

## Mapping evolutionary lineages

We compared the diversity of lineages of *Geophagus sensu stricto* with the biogeographic units proposed for fish in the Amazon basin. To understand the distribution of *Geophagus*

*sensu stricto* lineages in the Brazilian Amazon basin, we used the bGMYC result obtained from the Single-Locus Species Discovery (SLSD) methods and plotted in the biogeographic units proposed by *Dagosta & De Pinna, 2017* (Figs. S2–S6), the names of these units are also present in the delimitation tree (Fig. 2). While we did not favor any of the four methods *a priori*, bGMYC same as mPTP is relatively conservative, but at the same time it also captured common biogeographic patterns (*e.g.*, endemic taxa in the Araguaia-Tocantins basin) and delimited all described species, and therefore we used the bGMYC results for mapping. The visualization of the results has the objective of showing the spatial distribution of lineages, drainages and/or biogeographic units that shelter an elevated diversity of lineages, endemic lineages and those that are distributed in more than one biogeographic unit.

## RESULTS

We obtained 315 partial COI sequences of *Geophagus* from 72 localities in the Amazon basin. The addition of sequence data from GenBank and BOLD increased this dataset to 337 specimens from 77 localities inside and outside the Amazon basin. This alignment was then reduced to a total of 125 unique haplotypes from the *Geophagus sensu stricto* species group plus two '*Geophagus*' *steindachneri* haplotypes as outgroup. Sequence length varied from 317 to 702 bp, with a mean sequence length of 644 bp; 183 sites were variable. A total of 16 morpho-species were analyzed (10 nominal and 6 morphologically distinct but undescribed species). The morpho-species identified to the species level were: *G. altifrons* Heckel, 1840; *G. argyrostictus* Kullander, 1991; *G. dicrozoster* López-Fernández & Taphorn, 2004; *G. harreri* Gosse, 1976; *G. megasema* Heckel, 1840; *G. mirabilis* Deprá et al., 2014; *G. proximus* (Castelnau, 1855); *G. sveni* Lucinda, Lucena & Assis, 2010; *G. winemilleri* López-Fernández & Taphorn, 2004 and '*Geophagus*' *steindachneri* (Eigenmann & Hildebrand, 1922). Candidate species were identified as *Geophagus* sp. 1 to sp. 6 (Table 1).

The number of individuals per morpho-species varied from 1 to 104; haplotypes per morpho-species varied from 1 to 33; the number of drainage basins in which the morpho-species occur varied from 1 to 22; the number of sampled localities for each morpho-species varied from 1 to 39; the maximum intraspecific p-distance within morpho-species varied from 0 to 2.3%; the minimum interspecific p-distance between morpho-species varied from 0.6% to 13%. The locMin analysis optimized a divergence threshold of 1.86% (p-distance) for the dataset. For the 16 morpho-species identified *a priori*, fourteen (87.5%) were monophyletic and two (12.5%) were represented by a single haplotype (singleton). The most widely distributed nominal species had the greatest haplotype sharing. In *G. proximus* 40 individuals shared the same haplotype and *G. altifrons* 34 individuals shared the same haplotype (Fig. 2).

The number of species/lineages discovered by each method were 15 (mPTP), 14 (locMin), 18 (bGMYC), and 30 (GMYC) (Table 1). Species/lineages delimited by all four methods were: *Geophagus argyrostictus*, *G. dicrozoster*, *G. harreri*, *G. mirabilis*, '*G.*' *steindachneri*, *G. sveni*, *Geophagus* sp. 2, *Geophagus* sp. 4, and *Geophagus* sp. 6. Three of the four methods (mPTP, LocMin, and bGMYC) delimited *Geophagus* sp. 3 and

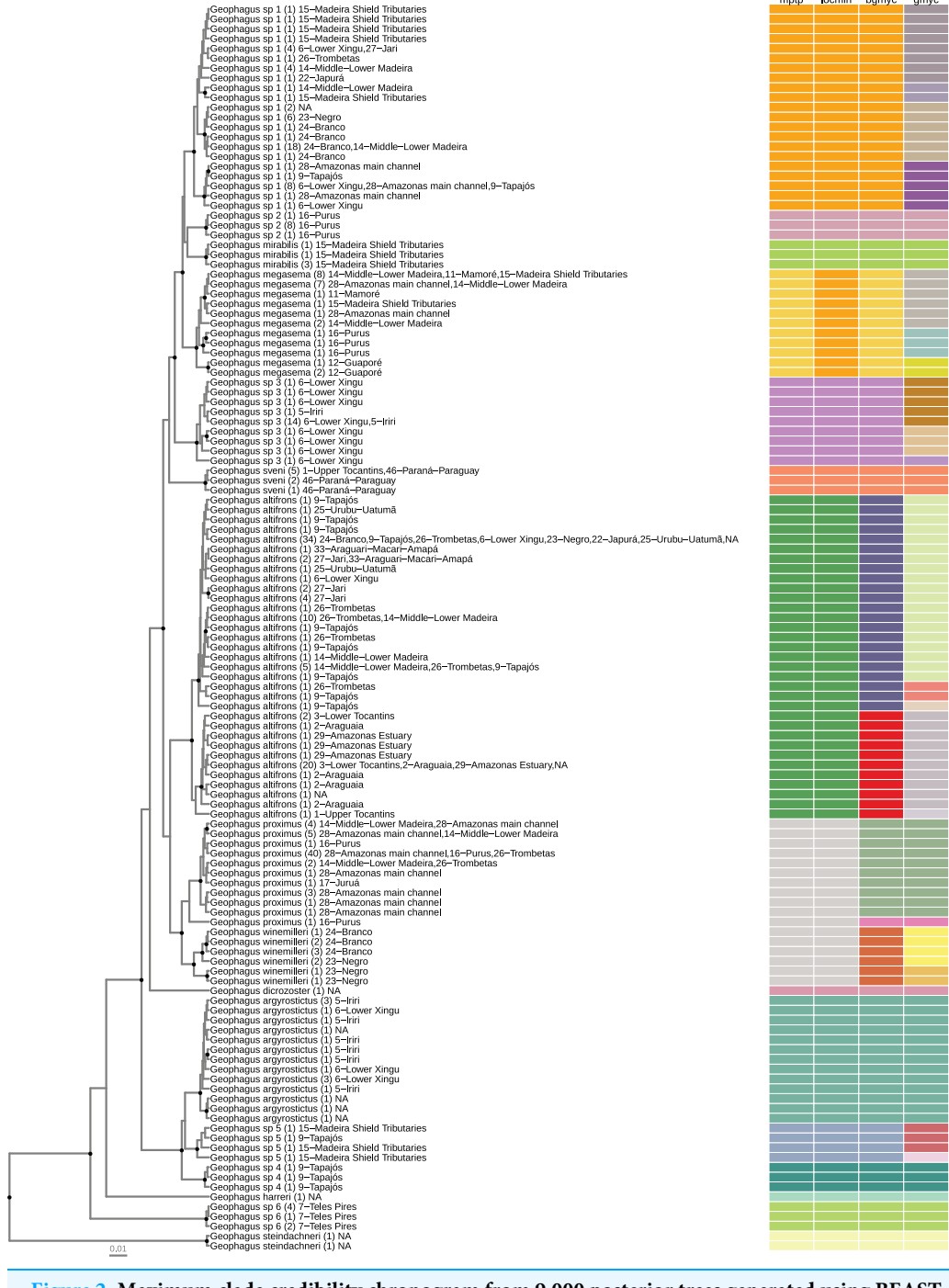

**Figure 2 Maximum clade credibility chronogram from 9,000 posterior trees generated using BEAST 2.6.** Dataset comprised 127 unique haplotypes (from a total 337) of *Geophagus* COI sequences. Bayesian posterior probabilities above 0.95 are shown as dark nodes. Species delimitations are shown by method as colored boxes. The number of collapsed individuals is indicated in parentheses and outside of it the locations where they were sampled. The tree was constructed in R 4.0.0 using the package 'ggtree' and the final graphic in Inkscape.

**Table 1 Summary statistics of the morpho-species analyzed in this study, including nominal species and candidate species (*Geophagus* sp.).**

| species | indivs | nHaps | drainages | localities | maxIntraDist | minInterDist | monophyly | gmyc | bgmyc | mptp | locmin |
|---------|--------|-------|-----------|------------|--------------|--------------|-----------|------|-------|------|--------|
| *Geophagus altifrons* | 104 | 33 | 22 | 39 | 0.023 | 0.021 | True | 5 | 2 | 1 | 1 |
| *Geophagus argyrostictus* | 17 | 13 | 2 | 3 | 0.009 | 0.021 | True | 1 | 1 | 1 | 1 |
| *Geophagus dicrozoster* | 1 | 1 | 1 | 1 | 0 | 0.044 | Singleton | 1 | 1 | 1 | 1 |
| *Geophagus harreri* | 1 | 1 | 1 | 1 | 0 | 0.079 | Singleton | 1 | 1 | 1 | 1 |
| *Geophagus megasema* | 26 | 11 | 8 | 9 | 0.022 | 0.012 | True | 3 | 1 | 1 | 1 |
| *Geophagus mirabilis* | 5 | 3 | 1 | 1 | 0.003 | 0.023 | True | 1 | 1 | 1 | 1 |
| *Geophagus proximus* | 60 | 11 | 9 | 11 | 0.021 | 0.006 | True | 2 | 2 | 1 | 1 |
| *Geophagus* sp. 1 | 57 | 21 | 13 | 22 | 0.019 | 0.012 | True | 4 | 1 | 1 | 1 |
| *Geophagus* sp. 2 | 10 | 3 | 1 | 2 | 0.002 | 0.021 | True | 1 | 1 | 1 | 1 |
| *Geophagus* sp. 3 | 22 | 9 | 3 | 5 | 0.013 | 0.025 | True | 3 | 1 | 1 | 1 |
| *Geophagus* sp. 4 | 3 | 3 | 1 | 1 | 0.005 | 0.023 | True | 1 | 1 | 1 | 1 |
| *Geophagus* sp. 5 | 4 | 4 | 3 | 3 | 0.013 | 0.021 | True | 2 | 1 | 1 | 1 |
| *Geophagus* sp. 6 | 7 | 3 | 1 | 1 | 0.002 | 0.085 | True | 1 | 1 | 1 | 1 |
| *Geophagus steindachneri* | 2 | 2 | 1 | 1 | 0.002 | 0.13 | True | 1 | 1 | 1 | 1 |
| *Geophagus sveni* | 8 | 3 | 3 | 5 | 0.004 | 0.029 | True | 1 | 1 | 1 | 1 |
| *Geophagus winemilleri* | 10 | 6 | 2 | 5 | 0.018 | 0.006 | True | 2 | 1 | 1 | 1 |

**Note:**
From the left to the right: number of individuals, number of haplotypes, number of sampled major drainages, number of sampled localities, maximum intraspecific divergence (p-distance), minimum interspecific divergence (p-distance), monophyly, and number of delimited clusters by each method (GMYC, bGMYC, mPTP, LocMin).

*Geophagus* sp. 5. Two of the four methods delimited *G. altifrons* (mPTP, LocMin), *G. megasema* (mPTP, bGMYC), *G. proximus* (bGMYC, GMYC), and *Geophagus* sp. 1 (mPTP, bGMYC). The only method to delimit *G. winemilleri* as a species/lineage was bGMYC.

The GMYC method indicated several more species/lineages within each clade than the other methods. None of the four methods were able to delimit *Geophagus surinamensis* and *G. proximus* sequences obtained from GenBank and BOLD databases as putative species. These sequences were nested within the species *Geophagus altifrons* and *G. sveni*, respectively.

## DISCUSSION

Our results unambiguously support the monophyly of the Amazonian species of *Geophagus*. With exception of '*Geophagus*' *steindachneri*–a trans-Andean species and the outgroup of our analyses–all the morpho-species sampled in this study belong to the *Geophagus sensu stricto* species group and were highly supported as monophyletic. The monophyly of the sensu *stricto* species group has been demonstrated elsewhere (*Farias et al., 1999*; *Farias, Ortí & Meyer, 2000*; *López-Fernández, Honeycutt & Winemiller, 2005*; *Smith, Chakrabarty & Sparks, 2008*), but relationships between all nominal species in this group have not yet been resolved.

After extensive sampling in the main channel of the Amazon basin and in most of its tributaries, our analysis covered a total of 10 nominal species of *Geophagus* out of a total of 20 species described for the sensu *stricto* species group. Additionally, we identified six

candidate species that could not be assigned to any other species of the *Geophagus sensu stricto* species group after the usage of original keys and descriptions of the species of the genus. The phylogenetic reconstruction and the SLSD methods delimited all six candidate species as independent and reciprocally monophyletic lineages (Fig. 2).

The maximum credibility tree recovered a highly supported clade containing six nominal species of the *Geophagus surinamensis* "complex" of *López-Fernández & Taphorn (2004)* (Fig. 2). The phylogenetic position of *G. dicrozoster*, however, was not fully resolved. Although this species is sister to the *G. surinamensis* "complex", this placement has low posterior probability support and is best interpreted as a polytomy which also includes the highly supported *G. argyrostictus* clade, comprised of *G. argyrostictus*, *Geophagus* sp. 4 and *Geophagus* sp. 5. At the base of the tree, the position of *G. harreri* was uncertain and had low posterior support. The species of *Geophagus sensu stricto* phylofenetically closest to '*G.*' *steindachneri*, outgroup in our analyses, was *Geophagus* sp. 6 from Azul River, a tributary of Teles Pires River.

## Unrecognized and misrepresented diversity

Our results show high diversity of *Geophagus* within the Amazon basin. The most conservative SLSD method (mPTP) indicates 15 species/lineages and the least conservative method (GMYC), 30 species/lineages (including '*Geophagus*' *steindachneri*). This last method usually tends to perform poorly and oversplits lineages when effective population sizes (Ne) and speciation rates are high (*Fujisawa & Barraclough, 2013*; *Dellicour & Flot, 2018*). Population structure and low sampling effort also affect results (*Papadopoulou et al., 2008*; *Lohse, 2009*), although it does not invalidate the method *per se* (*Papadopoulou et al., 2009*).

The vast majority of these species/lineages inhabit tributaries instead of the main channel of the Amazon River (Fig. S6). We found five of the six undescribed species occurring only in the tributaries, being restricted to one or few water bodies: *Geophagus* sp. 2 (Purus), *Geophagus* sp. 3 (Xingu), *Geophagus* sp. 4 (Tapajos), *Geophagus* sp. 5 (Madeira and Tapajos), and *Geophagus* sp. 6 (Teles Pires). The undescribed species *Geophagus* sp. 1 occurs in multiple tributaries (Branco, Japura, Jari, Madeira, Negro, Tapajos, Trombetas, and Xingu) and is also found in the main channel of the Amazon River.

The two and three individuals deposited in Genbank and BOLD under the epithet *Geophagus surinamensis* have common haplotypes of the geographically widespread *G. altifrons*. Given that either no vouchers exist or that the specimens are from "petshop" and of unknown geographic origin, and that *G. surinamensis* and *G. altifrons* are differentiated by subtle morphological differences (*Soares et al., 2008*), we considered these five samples misidentified *G. altifrons*.

The sequences in GenBank of individuals from the Paraná River basin identified as *G.* "*proximus*" (5) and as *G. sveni* from Paraná River basin and the Tocantins River (2) were delimited by the four methods as a clade (Fig. 2). *Benitez et al. (2018)* found minimal or no genetic divergence between these individuals and suggested that all specimens were *G. sveni* misidentified as *G.* "*proximus*". Furthermore, specimens collected

in the Paraná River basin and initially identified as *G.* cf. *proximus* (*Graça & Pavanelli, 2007*; *Sampaio & Goulart, 2011*) were later reassigned to *G. sveni* (*Ota et al., 2018*). These species share characteristics such as the presence of prominent mid-lateral black markings and absence of a complete infraorbital stripe, but differ in the markings in the preopercle, which is present in *G. proximus* and absent in *G. sveni*. Although these species also differ in the lateral bars that are present in *G. sveni* (*Lucinda, Lucena & Assis, 2010*) and absent in *G. proximus* (*López-Fernández & Taphorn, 2004*) in preserved specimens, discrimination of live individuals is not trivial. Both *G. proximus* and *G. sveni* are invasive species in the Paraná-Paraguay River basin, and their occurrence may be associated with the aquarium trade (*Gois et al., 2015*; *Ota et al., 2018*).

The most complicated case of incorrect species identification in our dataset lies within the *Geophagus* sp. 1 clade. Several individuals of this clade (posterior probability = 0.999) were incorrectly identified by their collectors during field expeditions as *Geophagus* cf. *winemilleri* (21 individuals), *Geophagus winemilleri* (six individuals), *Geophagus* cf. *abalios* (five individuals), *Geophagus* aff. *winemilleri* (four individuals), *Geophagus* aff. *altifrons* (four individuals), *Geophagus dicrozoster* (two individuals), *Geophagus abalios*, (one individual), *Geophagus* cf. *altifrons* (one individual), *Geophagus megasema* (one individual) and 12 more individuals identified as *Geophagus* sp. (Tapajós-Xingu) totaling 57 questionably identified individuals. All individuals of this clade possess five ventrally-inclined lateral bars, the first four being bisected by a clearer area, giving the appearance of two thinner bars, (as in *G. abalios*)–and the last one solid; the presence of midlateral spot (MLS) located in the third lateral bar; absence of infraorbital stripe (IOS); and absence of preopercular marks (POM). This combination of characters does not match character states of any nominal species, and therefore we classified these specimens as *Geophagus* sp. 1. *Geophagus* sp. 1 is also the third species–in addition to *G. altifrons* and *G. proximus*–to have a basin-wide distribution.

During reexamination of collected specimens, using either preserved specimens and/or photographs of the live specimens collected in the field, we also observed that both *G. altifrons* and *G. proximus* appear to be treated as "default species" by collectors. Many incorrect identifications have been attributed to these two species, perhaps because they are the most abundant and widely distributed species and/or the collection site was taken into account for species assignment.

## Geographic distribution of *Geophagus* lineages

The geographic distribution of the *Geophagus sensu stricto* lineages found in this study adds a new layer of complexity to understanding the evolutionary history of this species group in the Amazon basin. There are widely spread lineages in the Amazon River and its tributaries (*G. altifrons* sensu *stricto*, *G. proximus* sensu *stricto*, and the *Geophagus* sp. 1), lineages occurring in one or few tributaries (*G. altifrons* Araguaia-Tocantins, *G. megasema*, *G. winemilleri* and *Geophagus* sp. 5), and lineages restricted to a single or few geographically close localities (*G. argyrostictus*, *G. mirabilis*, *G. proximus* Purus, *Geophagus* sp. 2, *Geophagus* sp. 3, *Geophagus* sp. 4, *Geophagus* sp. 6 and *Geophagus sveni*) (Fig. 1).

Within the two broadly distributed described species, we delimited two lineages of *G. altifrons* (Fig. S2) and *G. proximus* (Fig. S3), one broadly distributed, one restricted. The restriced lineage of *G. altifrons* occurs in the Araguaia-Tocantins River basin and of *G. proximus* in the lower Purus River basin. The third widely distributed species (*Geophagus* sp. 1) appears to be a single lineage (Fig. S4).

*Geophagus altifrons* is a species with an eastern Amazon distribution pattern according to *Dagosta & De Pinna (2019)*. While most of our samples and collecting sites of both lineages of *G. altifrons* were to the east of the Purus Arch and thus would appear to be concordant with the hypothesis of *Dagosta & De Pinna (2019)* that the Purus Arch limits the distribution of *G. altifrons* to the east, *G. altifrons* sensu *stricto* also occurs in the Japura River, a western tributary of the Amazon, an indication that the Purus Arch does not limit its distribution.

*Kullander (1986*, *2003)* proposed that *G. proximus* occurs along the Ucayali-Solimões-Amazonas River to at least the Trombetas River. *Dagosta & De Pinna (2019)* report an even wider distribution for the species, including drainages of the Araguaia-Tocantins, Xingu, Madeira, Purus, Tefé, Ucayali, Japura, Negro, Branco, and Trombetas river basins, *i.e.*, extending the distribution of *G. proximus* to the mouth of the Amazon River and affluents. We are unable to confirm this broader distribution proposed by *Dagosta & De Pinna (2019)* since we detected *G. proximus* from the Jurua River east to the Tapajos River–the Tapajos River is a southern affluent of the Amazon shortly after its confluence with the Trombetas River. We also found a distinct lineage of *G. proximus* delimited by both GMYC and bGMYC methods (Fig. 2) that occurs in the middle Purus River (*G. proximus* Purus).

*Geophagus megasema* was found in the Madeira River basin downstream to Manicoré (Fig. S5). Three individuals from Ipixuna River–a middle Purus River tributary that at its headwaters is connected to the Madeira River–which were morphologically identified as *G.* cf. *altifrons* in the field–were nested within the *G. megasema* clade. There were also five individuals from Mamirauá Sustainable Development Reserve that were included within this clade. *Kullander (2003)* reports the distribution of the species in the Guapore River basin. *Dagosta & De Pinna (2019)* extends its distribution to the Beni-Madre de Dios River basin in Bolivia, the middle and lower Madeira River, and Japura River basin. Our results agree with this proposed distribution and extend the distribution of *G. megasema* to at least the middle Purus River. This sharing of fish fauna between these two basins was reported in the Madeira-Purus interfluvial checklist (*Barros et al., 2011*, *2013*).

*Geophagus winemilleri* was delimited by two methods (bGMYC and GMYC) as lineages different from *G. proximus*. *Geophagus winemilleri* has a known distribution in the Negro and Orinoco river basins (*López-Fernández & Taphorn, 2004*), but we have detected this species in the Branco River as well (Fig. S5). Despite the geomorphological and the physical-chemical differences between these rivers, other studies have already demonstrated the sharing of the ichthyofauna between them (*Dagosta & De Pinna, 2019*).

The lineage *Geophagus* sp. 5 occurs in the clear water rivers of the Aripuana, Machado, and lower Tapajos (Fig. S6). Madeira and Tapajos are neighboring basins and both

Aripuana and Machado are tributaries of the Madeira River that drain the Brazilian Shield. There are several records of faunal sharing between these basins, mostly between the Aripuana, Guapore, and Machado rivers with the Juruena River, a Tapajós River tributary (*Dagosta & De Pinna, 2019*). The sharing of ichthyofauna between these basins can be mostly explained by geomorphological processes that resulted in stream capture events across the Madeira-Tapajos interfluvial, which in turn resulted in geodispersal of entire faunas (*Dagosta & De Pinna, 2019*).

The endemic species *G. argyrostictus* was sampled in the Iriri and Xingu rivers (Fig. S6). The species is only known from the upper and middle reaches of the Xingu River (*Kullander, 1991*), although the individuals of the upper Xingu (Cachoeira von Martius) analyzed by *Kullander (1991)* differ slightly from the middle Xingu (Altamira, Belo Monte, and Cachoeira do Espelho) in morphometric measurements and in average counts of gill rakers, pectoral fin rays, and abdominal vertebrae. However, we have no data to either confirm or to reject that the specimens from the upper and middle Xingu River represent distinct lineages. *Kullander (1991)* also states that the only other *Geophagus* species in the Xingu River is *G. altifrons*, which is sympatric with *G. argyrostictus*. However, in addition to *G. argyrostictus* and *G. altifrons* our results suggest the occurrence of two other lineages in the Xingu River basin: *Geophagus* sp.1, the broadly distributed species which occurs in the lower Xingu River, and *Geophagus* sp. 3 an apparent endemic of the middle Xingu region (Fig. 1). *Geophagus* sp. 3 also occurs in the Iriri and Bacaja rivers, both tributaries of the middle Xingu River.

*Geophagus mirabilis* was found at a single locality in the Aripuana River, downstream of the Dardanelos falls, and *G. sveni* was collected only in the upper Tocantins River. Both rivers are recognized for having a large number of endemic species (*Lucinda, Lucena & Assis, 2010*; *Deprá et al., 2014*).

The lineage *Geophagus* sp. 2 occurs only in the Purus River (Fig. S6). The upper Purus River has already been recognized in several studies as a region that possesses structured fish populations, and events in the geological evolution of this region, such as the elevation of the Fitzcarrald Arch and drainage captures are suggested as responsible for these structured populations (*Machado et al., 2017*; *Santos, Hrbek & Farias, 2018*). *Geophagus* sp. 4 was found only in the region of Vila Pimental, Tapajos River, and *Geophagus* sp. 6 was found occurring only in the Azul River, a tributary of the Teles Pires River (Fig. S6). Tapajós and Xingu rivers are recognized for elevated levels of ichthyofaunal endemism (*Fitzgerald et al., 2018*; *Oberdorff et al., 2019*). Recent studies have only increased the number of endemics in these basins (*Silva-Oliveira, Canto & Ribeiro, 2015*, *2016*; *Carvalho, 2016*; *Fitzgerald et al., 2018*).

## Implications for conservation

The discovery of possible new species of cichlid fishes in the Amazon basin is yet another example of how little we know of the biodiversity around us. Lineages are products of evolutionary processes (*De Queiroz, 2007*). When conserving these lineages, the processes which generated them are also conserved (*Coates, Byrne & Moritz, 2018*; *Hrbek et al., 2018*). Floodplains, rapids, and waterfalls are in constant threat due to human activities

(*Castello et al., 2013*; *Castello & Macedo, 2016*). When these areas are permanently altered by humans, not only ichthyofaunal diversity is lost, the entire evolutionary history recorded in these species is lost as well, hindering our efforts to understand the processes that generated the astonishing diversity of Amazonian fishes. Habitat destruction, deforestation, mining, and hydroelectric dam construction are examples of human activities that harm both hydrological connectivity and ecosystem services (*Castello et al., 2013*; *Castello & Macedo, 2016*).

Many of the new lineages discovered in our study were found in areas near rapids or waterfalls–environments commonly preferred for the construction of hydroelectric dams (*Winemiller et al., 2016*). One example is the candidate species *Geophagus* sp 4. which was found only in the Tapajós River, in the region of Vila Pimental, location planned for the construction of the São Luiz do Tapajós hydroelectric plant. The construction and filling of the dam leads to the disappearance of these environments. Therefore, the endemic species sheltered in these singular environments may disappear if the areas in which they occur are permanently altered. Thus, the recognition of these lineages as conservation dependent is important to guide conservation actions in these impacted areas and to direct efforts to the formal examination of these lineages as putative species, which could already be threatened or in danger of extinction.

## CONCLUSIONS

The results of the single locus species delimitation methods complemented morphological delimitations; all six morphospecies identified in this study were also delimited as species/lineages, and all described species with the exception of *G. winemilleri* and *G. proximus* in the case of mPTP and bGMYC, were delimited as lineages or clusters of species. These results therefore provide strong and consistent evidence for additional taxonomic diversity in this group. Although not formally described in this study, these six new species increase the taxonomic diversity of *Geophagus sensu stricto* by 30%. Of the six new species, five are endemic or inhabit areas subject to major human threats. Our sampling was not exhaustive, however, and we expect that additional species will be discovered principally in poorly sampled regions of the Amazon basin.

## ACKNOWLEDGEMENTS

We are grateful to all the volunteers/colleagues who collected field samples and who for years have contributed to the tissue collection (CTGA/UFAM). We especially thank Joiciane Farias, Natasha Meliciano, Mário Nunes, Rupert Collins, Daniel Toffoli, Stuart Willis, and Hernán López for their help with field collection, Carolina Doria (Universidade Federal de Rondônia-UNIR, Coleção de tecidos do laboratório de Ictiologia e Pesca), Camila Ribas (Instituto Nacional de Pesquisas da Amazônia-INPA, Coleção de Recursos Genéticos Animais), Lúcia Py-Daniel (INPA, Coleção Ictiológica) for the tissue samples from the ichthyological collections. This study is part of AMX's doctoral dissertation in the Genetics, Conservation and Evolutionary Biology graduate program INPA/UFAM.

### Funding

This study was financed by following grants: Conselho Nacional de Desenvolvimento Científico e Tecnológico (CNPq/SISBIOTA-BioPHAM) (CNPq grant no 563348/2010) and Fundação de Amparo a Pesquisa do Estado do Amazonas (FAPEAM/SISBIOTA) awarded to Izeni Pires Farias. Aline Mourão Ximenes was supported by Coordenação de Aperfeiçoamento de Pessoal de Nível Superior-Brasil (CAPES) doctoral fellowship, financial code 001. Publication costs were supported by Fundação de Amparo á Pesquisa do Amazonas (FAPEAM/PAPAC grant No. 005/2019). Additional logistical support were also by Laboratório de Evolução e Genética Animal (LEGAL/UFAM), Coleção de Tecidos de Genética Animal (CTGA/UFAM), Ichthyological collection of the Universidade Federal de Rondônia (UNIR) and Ichthyological collection of the Instituto Nacional de Pesquisas da Amazônia (INPA). The funders had no role in study design, data collection and analysis, decision to publish, or preparation of the manuscript.

### Grant Disclosures

The following grant information was disclosed by the authors:
Conselho Nacional de Desenvolvimento Científico e Tecnológico (CNPq/SISBIOTA-BioPHAM): 563348/2010.
Fundação de Amparo a Pesquisa do Estado do Amazonas (FAPEAM/SISBIOTA, FAPEAM/PAPAC): 005/2019.
Coordenação de Aperfeiçoamento de Pessoal de Nível Superior- Brasil (CAPES): 001.
Laboratório de Evolução e Genética Animal (LEGAL/UFAM).
Coleção de Tecidos de Genética Animal (CTGA/UFAM).
Universidade Federal de Rondônia (UNIR).
Instituto Nacional de Pesquisas da Amazônia (INPA).

### Competing Interests

Tomas Hrbek is an Academic Editor for PeerJ. All other authors declare that there are no competing interests.

### Author Contributions

- Aline Mourão Ximenes conceived and designed the experiments, performed the experiments, analyzed the data, prepared figures and/or tables, authored or reviewed drafts of the paper, and approved the final draft.
- Pedro Senna Bittencourt performed the experiments, analyzed the data, prepared figures and/or tables, authored or reviewed drafts of the paper, and approved the final draft.
- Valéria Nogueira Machado analyzed the data, authored or reviewed drafts of the paper, and approved the final draft.
- Tomas Hrbek analyzed the data, authored or reviewed drafts of the paper, and approved the final draft.

Peer⌡

- Izeni Pires Farias conceived and designed the experiments, performed the experiments, analyzed the data, authored or reviewed drafts of the paper, and approved the final draft.

## Animal Ethics

The following information was supplied relating to ethical approvals (*i.e.*, approving body and any reference numbers):

All individuals were captured and sampled under license granted by the Instituto Brasileiro do Meio Ambiente e dos Recursos Naturais Renováveis (IBAMA/SISBIO permit #62216-1). Collection of organisms was undertaken in accordance with the ethical recommendations of the Conselho Federal de Biologia (CFBio; Federal Council of Biologists), Resolution 301 (December 8, 2012).

## Data Availability

All sequences, unique haplotypes used for species discovery methods, the number of genbank hits and the metadata for all sequences used in this study are available in the Supplementary Files.

The sequences are available at GenBank: MZ504295–MZ504609.

## Supplemental Information

Supplemental information for this article can be found online at http://dx.doi.org/10.7717/peerj.12443#supplemental-information.

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
