# Peer review of "Mapping the hidden diversity of the Geophagus sensu stricto species group (Cichlidae: Geophagini) from the Amazon basin"

_PeerJ, doi:10.7717/peerj.12443_

## Round 0.1 · original submission · Minor Revisions

I have now received three reviewer reports for your study, two of which recommend acceptance and one recommending a minor revision. I agree with the latter that already admirable paper will benefit from a closer scrutiny for clarity, formatting and syntax.

Please attend to the following.
1. Place the article in peerJ template, including, preferably a structured abstract - template can be found here. https://peerj.com/new/
2. The paper can benefit from a closer scrutiny for syntactical errors. Consider breaking down complicated sentences into simple ones that can be easily understood by a more diverse readership.
3. Attend to the points regarding clarity and syntax raised by the reviewer.
4. Though not essential, explicitly mentioning the species concepts you follow especially with the species delimitation analysis would help the reader understand your paper in a broader context. For your paper, mainly the Morphological Species concept and Phylogenetic Species Concept (PSC) applies. But you may also want to mention General Lineage Method and its implementation in the context of integrative taxonomy when describing the lineages highlighted as new taxa in the future.
5. Note that peerJ allows a 500 word (3000 characters) Abstract, and I think you should make use of it to express your analyses, background, results and its implications fully. This is up to your discretion and not essential.

Reviewer 1 ·

Basic reporting

This study was well-designed and successfully executed, using modern community standards of molecular species delineation in combination with morphological data in Amazonian fishes. The main conclusion is well supported, for the existence of six morphologically distinct but as yet undescribed species of i/Geophagus/-i.

My only suggestion, which I’m sure the authors will agree, is to modify the last sentence of the Abstract to: “Our results highlight the importance of combining DNA and morphological data in biodiversity asessment studies especially in taxonomically diverse tropical biotas."

Experimental design

The research question, methods and interpretations are excellent, sufficient described, and meet community standard for the field.

Validity of the findings

The results are interpreted reasonably in light of the methods employed.

Reviewer 2 ·

Basic reporting

In this study by Ximenes et al. present an interesting and a well-written study on the species diversity of the cichlid group Geophagus in the Amazonian River basin in South America. The authors use an impressive dataset of mitochondrial cox1 marker and carryout several single locus species discovery methods to understand and report the diversity of this group. I have only minor comments which are directly commented on the pdf of the manuscript which is attached herewith.

Experimental design

no comments

Validity of the findings

no comments

Annotated reviews are not available for download in order to protect the identity of reviewers who chose to remain anonymous.

Reviewer 3 ·

Basic reporting

No comment. The paper was well done and had no such deficiencies.

Experimental design

No comment. The experimental design was good.

Validity of the findings

Findings were valid and clearly explained.

Additional comments

This paper treats species delimitation in the Cichlid genus Geophagus in the Amazon basin. The authors use a single locus species delimitation method approach based on the DNA barcoding gene COI to test species delimitation based on broad sampling and a relatively large number of samples of Geophagus. They also examined morphospecies from photos and museum specimens.

The overview of the problem and the approach they took is good, the references cited are appropriate and the methods are well done with the authors using several different sequence based methods to delimit the species. They find some described species cannot be justified as separate species, many misidentified specimens in collections, and several putative new species. The distribution of each species is presented in the Discussion. They end the paper discussing the conservation implications of recognizing putative undescribed species given the environmental problems plaguing the Amazon basin like dam construction.

I found the paper quite well done. It treats an important biological problem and makes an important contribution to moving knowledge of the genus forward. The paper will also likely serve as a valuable reference for other researchers wishing to conduct similar studies on other taxa.

---

## Round 0.2 · accepted · Accept

The authors have addressed the points raised by the reviewers and me to an appreciable level and hence the manuscript fulfills the editorial requirements for acceptance. I congratulate the authors for their rigorous sampling effort and a well-done study.